# Growing Transformers: Modular Composition and Layer-wise Expansion on a Frozen Substrate

## Abstract

The prevailing paradigm for scaling large language models (LLMs) involves monolithic, end-to-end training, a resource-intensive process that lacks flexibility. This paper explores an alternative, constructive scaling paradigm, enabled by the principle of emergent semantics in Transformers with frozen, non-semantic input embeddings. We posit that because high-level meaning is a compositional property of a Transformer's deep layers, not its input vectors, the embedding layer and trained lower layers can serve as a fixed foundation. This liberates backpropagation to focus solely on newly added components, making incremental growth viable. We operationalize this with a layer-wise constructive methodology that combines strict layer freezing in early stages with efficient, holistic fine-tuning of the entire model stack via low-rank adaptation (LoRA) as complexity increases. This method not only demonstrates stable convergence but also reveals a direct correlation between model depth and the emergence of complex reasoning abilities, such as those required for SQuAD, which are absent in shallower models. In a controlled study, our constructively grown model rivals the performance of a monolithically trained baseline of the same size, validating the efficiency and efficacy of the approach. Our findings suggest a path towards a paradigm shift from monolithic optimization towards a more biological or constructive model of AI development. This opens a path for more resource-efficient scaling, continual learning, and a more modular approach to building powerful AI systems. We release all code and models to facilitate further research.

## 1 INTRODUCTION

The pursuit of artificial general intelligence has led to the development of increasingly massive Large Language Models (LLMs). The dominant methodology for their creation is monolithic pre-training, where a model with a fixed, gargantuan architecture is trained end-to-end on vast datasets. While effective, this approach is computationally prohibitive, environmentally costly, and fundamentally inflexible. Once trained, modifying or extending such models without inducing catastrophic forgetting is a significant challenge.

A recent finding challenged a core assumption of this paradigm by demonstrating that Transformers can achieve robust reasoning performance using a completely frozen, non-semantic embedding layer (Bochkov, 2025). This established that semantic understanding is an emergent property of the Transformer's compositional architecture, not an inherent feature of its input vectors.

This paper investigates a direct architectural consequence of this discovery: if semantics emerge from depth, can we build powerful models by progressively increasing that depth, layer by layer, upon a stable, frozen foundation? We answer this question affirmatively by proposing and validating a "constructive learning" framework for scaling Transformers. We hypothesize that a frozen embedding layer acts as an immutable substrate, allowing deeper computational structures to be built and trained incrementally, a process that is far more resource-efficient than monolithic training.

To validate this hypothesis, we conduct two distinct sets of experiments:

- **Feasibility at Scale under Resource Constraints:** First, we demonstrate the methodology's viability for building models whose full optimizer state exceeds the memory of a single accelerator. We progressively grow a large-scale Transformer to a final size of 2.3B parameters, while keeping the number of trainable parameters at each growth stage within a fixed budget (approx. 740M, fitting on a single H100 GPU). The training throughput was scaled using data parallelism across additional accelerators. This experiment shows how complex capabilities emerge as the model grows from 1 to 6 layers, demonstrating a practical path to training models that would otherwise be infeasible to train monolithically on such hardware.

- **Controlled Comparative Study:** Second, to rigorously evaluate the performance of our approach, we conduct a controlled experiment on smaller (247.6M), 9-layer models. We compare our constructive method using frozen embeddings against three critical baselines: (a) a traditional, monolithically trained model of the same final architecture; (b) a constructively grown model using standard trainable embeddings; and (c) an ablation with a minimalist embedding composed of 16-component binary vectors to test the limits of the principle.

Our results show that this constructive approach is effective, offering a path towards more sustainable, adaptable, and interpretable model scaling. All code and models are released to foster further progress[1].

## 2 RELATED WORK

Our work builds upon several lines of research in neural network training and modularity, but offers a distinct synthesis enabled by our frozen-embedding foundation.

### 2.1 Greedy Layer-Wise Training

The idea of training deep networks one layer at a time was pioneered by Hinton et al. (Hinton et al., 2006) for Deep Belief Nets and explored by Bengio et al. (Bengio et al., 2007) for deep autoencoders. These methods were primarily used for pre-training to find a good initialization for subsequent end-to-end fine-tuning. Our approach differs fundamentally: we use progressive layer-wise training not for initialization, but as the primary, constructive training process itself, building upon a fixed, non-trainable base.

### 2.2 Progressive and Modular Architectures

Progressive Neural Networks (PNNs) (Rusu et al., 2016) achieve continual learning by adding new network "columns" for each new task. Adapter-based methods like AdapterFusion (Pfeiffer et al., 2021) inject small, trainable modules into a frozen base model. Our layer-wise growth is a form of vertical, rather than horizontal, expansion. The use of Low-Rank Adaptation (LoRA) (Hu et al., 2022) in our growth stages serves as an efficient tool for global readjustment (Schulman & Lab, 2025), but the core principle remains constructive and layer-wise.

The novelty of our work lies in demonstrating that a frozen, non-semantic representational substrate makes these techniques drastically simpler and more powerful in the context of modern Transformers.

## 3 CONSTRUCTIVE LEARNING METHODOLOGY

The cornerstone of this methodology is the frozen visual Unicode embedding layer, as detailed in (Bochkov, 2025). This provides a fixed, deterministic mapping from any token in a vocabulary to a $d_{\mathrm{model}}$-dimensional vector. This shared (V, $d_{\mathrm{model}}$) embedding matrix is the immutable foundation for all models in this study.

---

[1]The code to reproduce our experiments is provided in the supplementary materials.

### 3.1 Progressive Layer-Wise Growth

Instead of initializing a deep, N-layer Transformer and training it all at once, we "grow" it iteratively. The process, analogous to a locomotive pulling one wagon at a time, is as follows:

**Initialization:** Create a model $M_1$ with a single Transformer block ($n_{\text{layer}} = 1$) on top of the frozen embedding layer.

**Train Layer 1:** Train $M_1$ on the full corpus until convergence.

**Freeze and Stack:** Freeze the weights of the trained block in $M_1$. Add a new, randomly initialized Transformer block on top, creating model $M_2$.

**Train Layer 2:** Train $M_2$, where only the weights of the newly added second layer are trainable.

**Iterate:** Repeat this process, adding one layer at a time ($M_3$, $M_4$, ...), training only the newest layer at each step. This incremental process acts like a curriculum, with each new layer learning to compose the representations produced by the already-competent stack beneath it.

This layer-wise process can be strategically interspersed with holistic fine-tuning stages. For deeper models (e.g., after reaching 3 or 5 layers), we introduce an alternative training phase. Instead of adding a new layer, we freeze the entire stack and apply Low-Rank Adaptation (LoRA) (Hu et al., 2022) adapters to all existing layers.

Crucially, this switch in strategy operates under a **constant training budget**. The number of trainable parameters in the LoRA adapters is carefully chosen to match the number of parameters in a single new Transformer layer ($\approx$740M in our large-scale experiment). This allows us to reallocate the same computational resources from growing the model's depth to enhancing its internal coherence. This LoRA-based fine-tuning allows the entire network to adapt to its new depth, promoting global integration of knowledge and preventing the ossification of lower layers, all without increasing the instantaneous memory requirements for training.

## 4 EXPERIMENTAL SETUP

### 4.1 Overview of Experiments

To validate our constructive learning framework, we conduct two distinct sets of experiments. The first demonstrates the **feasibility** of our method for training large-scale models under typical hardware memory constraints. The second provides a **rigorous controlled comparison** of our approach against key baselines to evaluate its performance and isolate the contribution of frozen embeddings.

### 4.2 Model Architectures and Baselines

All models in our experiments are standard decoder-only Transformers, similar to GPT-2, using rotary embeddings and GELU activations.

#### 4.2.1 Large-Scale Feasibility Study

For the feasibility study, we use a large model configuration with a hidden dimension $d_{\text{model}} = 4096$, $n_{\text{head}} = 32$, and a vocabulary size of 131,072. The model, which we refer to as `ABS-BVV`, is grown progressively from 1 to 6 layers. At each stage, only the newly added layer(s) are trained, keeping the number of trainable parameters constant at approximately 740M. This ensures the optimizer states fit within the memory of a single 80GB H100 GPU. The final 6-layer model reaches 2.3B parameters.

### 4.2.2 Controlled Comparative Study

For the controlled study, we use a smaller, fixed architecture of 9 layers with $d_{\text{model}} = 1024$, $n_{\text{head}} = 32$, and a vocabulary size of 65,536, resulting in models of approximately 247.6M parameters. We compare four models:

- **Constructive Frozen ('Model UNICODE 1_9'):** Our proposed method. It is grown incrementally in three stages (layers 1-3, then 4-6, then 7-9). The visual Unicode embedding layer is frozen from the start, as are previously trained layers at each new stage. The final model has 247.6M total parameters, of which 209.9M are frozen.

- **Monolithic Trainable ('Model unfrozen (baseline)'):** The primary baseline. A standard 9-layer Transformer with a randomly initialized, fully trainable embedding layer. It is trained end-to-end in a traditional, monolithic fashion. All 247.6M parameters are trainable.

- **Monolithic Frozen ('Model frozen UNICODE (baseline)'):** The secondary baseline. A standard 9-layer Transformer with fully frozen and Unicode-predefined embedding layer. It is trained end-to-end in a traditional, monolithic fashion. 67.1M (of 247.6M) parameters are frozen.

- **Constructive Trainable (Ablation 1, 'Model unfrozen 1_9'):** This model isolates the effect of the growth strategy itself. It is grown identically to our proposed method (in three stages), but starts with a standard, trainable embedding layer. This tests whether constructive growth is effective without the frozen substrate.

- **Constructive Frozen (16-bit Binary, Ablation 2, 'Model 16_bit 1_9'):** This model tests the limits of the emergent semantics principle. It uses a minimalist, 16-dimensional binary vector for each token (derived from the token's integer ID) as its frozen embedding. This vector is projected to $d_{\text{model}} = 1024$ via simple repetition. The model is then grown constructively. The final model has 181.6M total parameters, of which 143.9M are frozen.

- **Monolithic Frozen ('Model frozen 16-bit (baseline)'):** The third baseline. A standard 9-layer Transformer with fully frozen 16-dimensional binary embedding layer. It is trained end-to-end in a traditional, monolithic fashion. 1.0M (of 181.6) parameters are frozen.

### 4.3 Training Datasets and Protocol

The training corpus for all experiments consists of a subset of English Wikipedia combined with SFT datasets (SQuADv2, CommonsenseQA, ARC), totaling approximately 4B tokens. Models are trained using the AdamW optimizer. For the constructive models, training proceeds in stages (e.g., layers 1-3 are trained until convergence, then frozen; then layers 4-6 are added and trained, etc.). For the monolithic baseline, all 9 layers are trained simultaneously from the start.

## 5 EXPERIMENTS AND RESULTS

### 5.1 Results: Progressive Layer-Wise Growth

We grew a model from 1 ('abs-bvv-1') to 6 layers ('abs-bvv-6'). Figure 1 illustrates the training process. Each sharp spike in the loss corresponds to the addition of a new, untrained layer, followed by rapid convergence, demonstrating the stability of the method.

The results, summarized in Figure 2, reveal a clear pattern. General reasoning ability on MMLU increases steadily with depth, from 18.08% with one layer to 21.63% with six. More strikingly, the ability to perform complex extractive question-answering (SQuAD) is virtually non-existent in shallow models (1.21% at $n_{\text{layer}} = 1$). A significant signal appears only at 'n_layer=3' (3.75%), and reaching its peak performance in our experiments at $n_{\text{layer}} = 6$ (5.55%). This compellingly demonstrates that complex capabilities are an emergent property of model depth. Figure 3 shows how performance on various MMLU subjects evolves as layers are added.

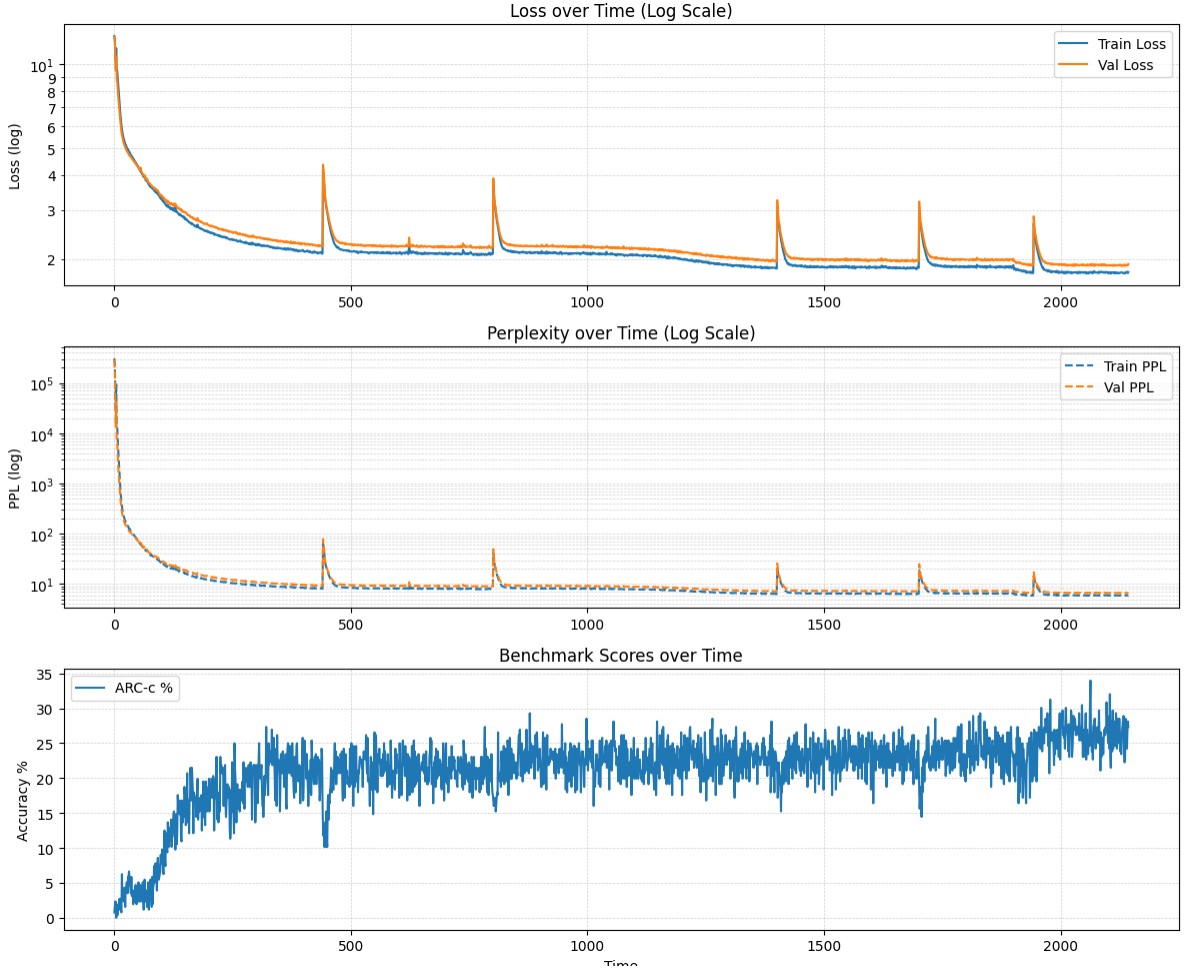

Figure 1: Training dynamics during progressive layer-wise growth. Each loss spike marks the stacking of a new layer, followed by rapid convergence. The ARC-c metric shows a corresponding increase in capability.

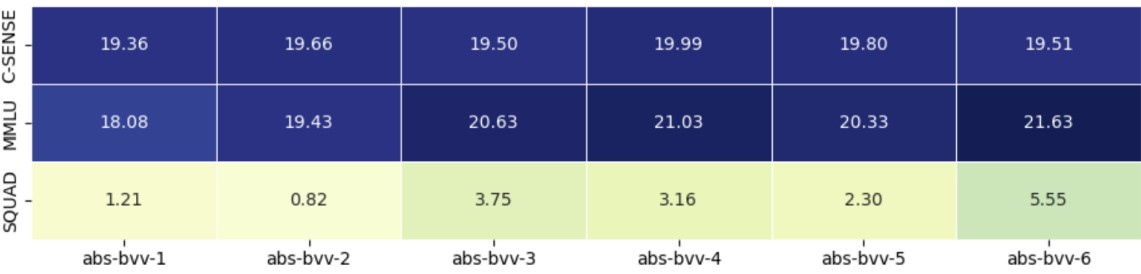

Figure 2: Benchmark performance as a function of model depth. Note the significant jump in SQuAD score at 'n_layer=3', indicating the emergence of complex reasoning.

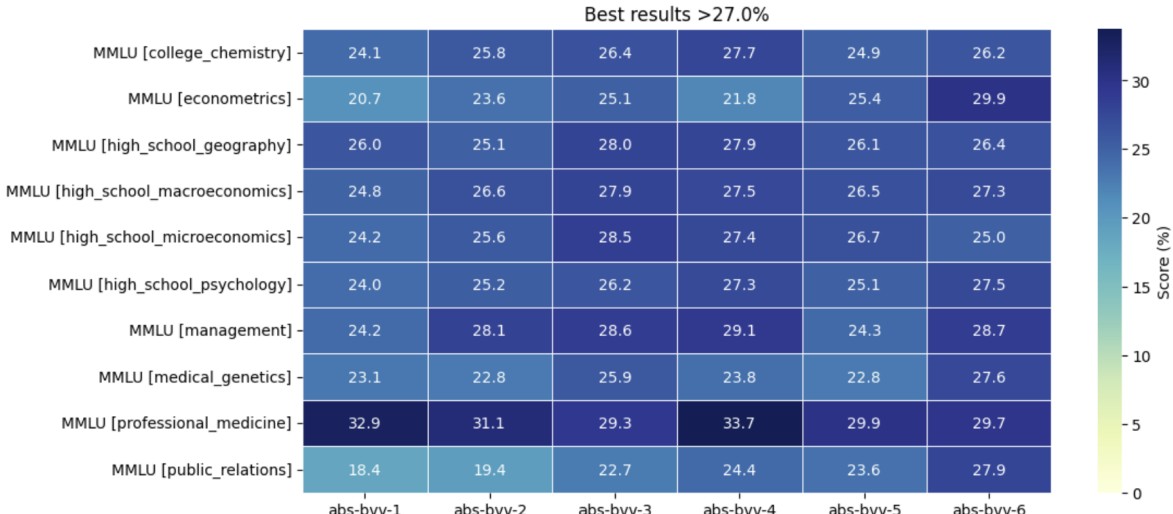

Figure 3: MMLU performance on select subjects as a function of model depth, illustrating how different reasoning capabilities strengthen as the model grows.

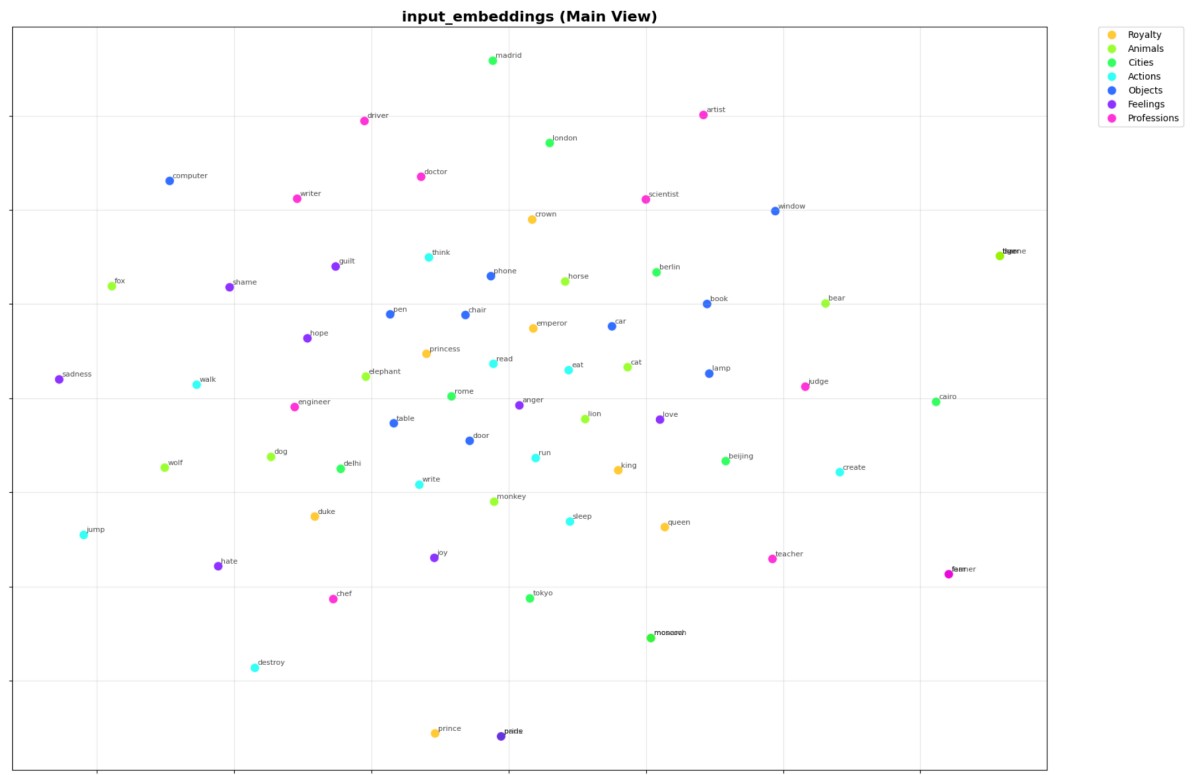

Figure 4: t-SNE visualization of frozen input token embeddings for 'abs-bvv-6'. In the this frozen embeddings, semantic groups (e.g., numbers, professions, animals) located randomly and without clustering. This fully corresponds to the initial condition of an absence of semantics in the visually precomputed embedding layer.

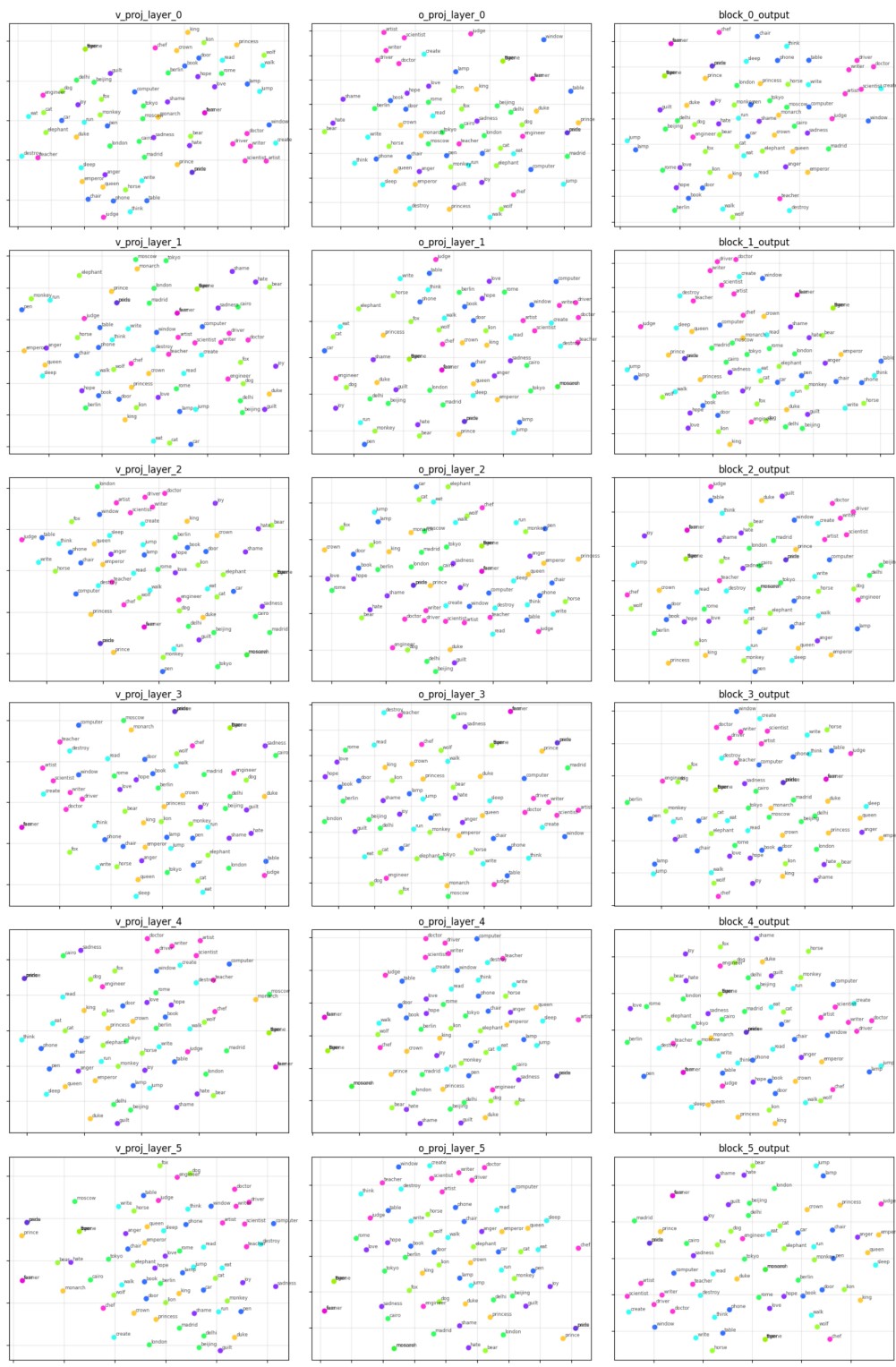

Figure 5: t-SNE visualization from layer 0 to layer 5 of attention projections (v_proj and o_proj), and transformer block outputs for 'abs-bvv-6' with frozen input embedding layer. In contrast to the input layer, clear semantic clusters emerge and sharpen with depth, demonstrating that meaning is constructed compositionally by the network.

## 5.2 Results: Controlled Comparative Study

The controlled comparative study was designed to rigorously evaluate our constructive learning method against traditional monolithic training and key ablations. The results reveal a clear trade-off between optimization on the training objective (loss) and performance on downstream reasoning tasks (MMLU).

As shown in Figure 6, all models exhibit stable convergence. The monolithic baseline, with all its parameters being trainable, unsurprisingly achieves the lowest final training loss. In contrast, the constructive models, particularly those with frozen components, plateau at a slightly higher loss. This is an expected outcome for two key reasons: first, a significant portion of their parameters are not directly optimized for the training objective; second, and critically, this controlled study intentionally omits the LoRA-based holistic tuning stages used in our large-scale experiment. This was done to strictly isolate the raw effect of the layer-stacking methodology.

The key finding of this study is presented in Figure 7. Our proposed 'Model UNICODE 1_9' outperforms the monolithically trained baseline on the MMLU benchmark, achieving a final score approximately 5 percentage points higher. This result supports hypothesis that decoupling structural and semantic representation allows the Transformer's deeper layers to specialize more effectively in high-level reasoning.

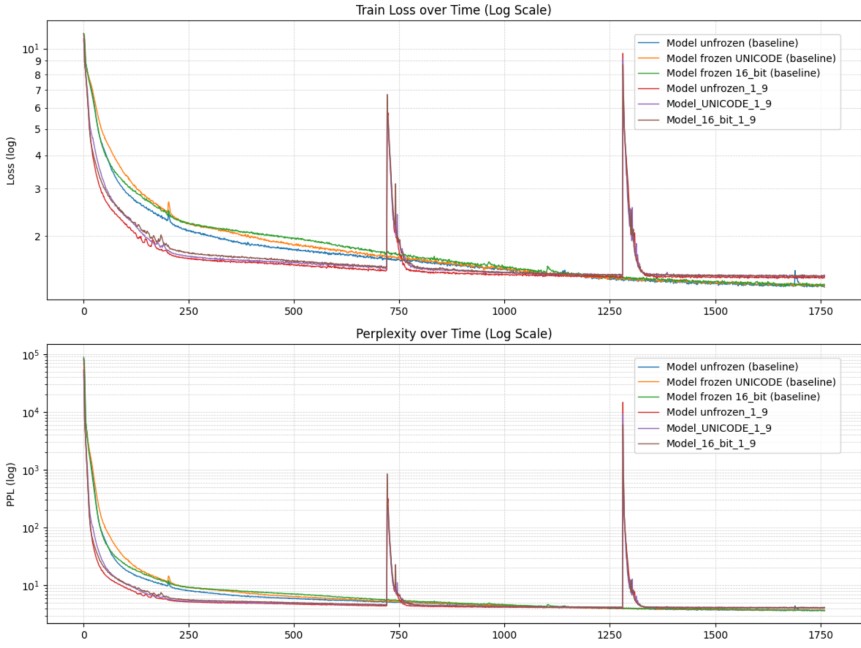

Figure 6: Training dynamics during progressive layer-wise growth for 'Model unfrozen (baseline)', 'Model frozen UNICODE (baseline)', 'Model frozen 16_bit (baseline)', 'Model UNICODE 1_9', 'Model unfrozen 1_9' and 'Model 16_bit 1_9'. Each loss spike marks the stacking of a new layer group, followed by rapid convergence.

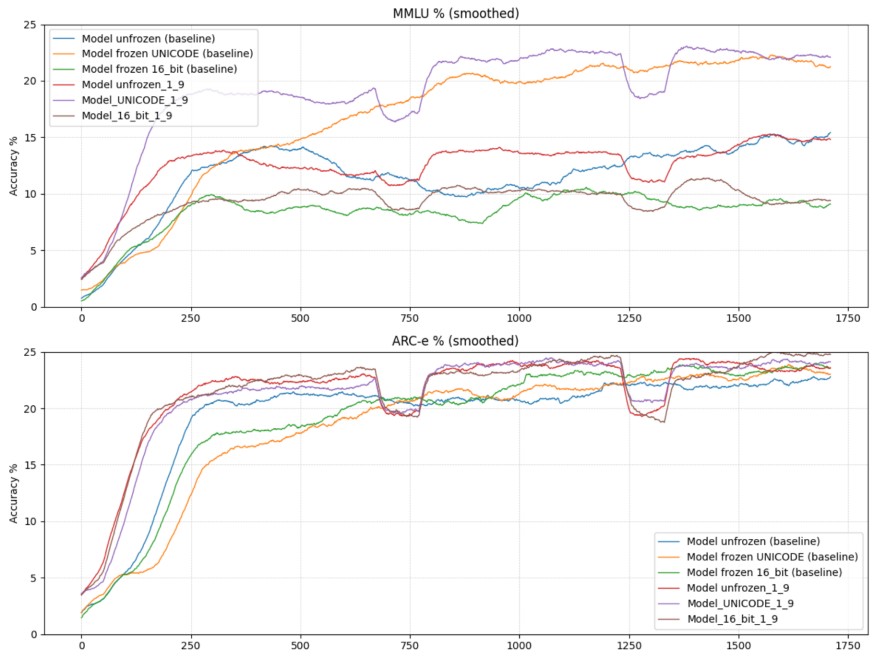

Figure 7: MMLU and ARC-E metric dynamics during progressive layer-wise growth for 'Model unfrozen (baseline)', 'Model frozen UNICODE (baseline)', 'Model frozen 16_bit (baseline)', 'Model UNICODE 1_9', 'Model unfrozen 1_9' and 'Model 16_bit 1_9'.

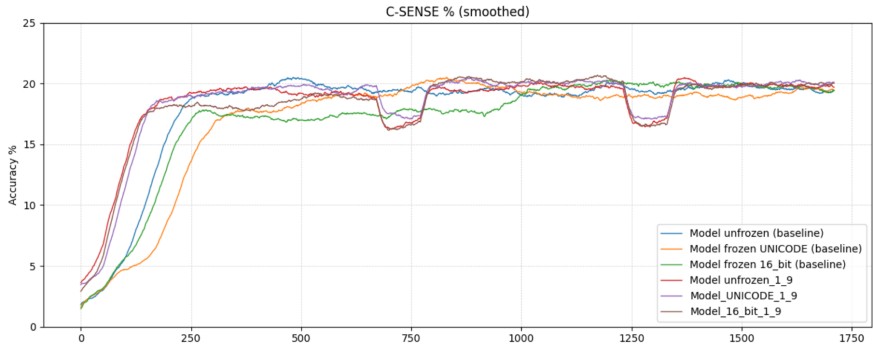

Figure 8: C-SENSE metric dynamics during progressive layer-wise growth for 'Model unfrozen (baseline)', 'Model frozen UNICODE (baseline)', 'Model frozen 16_bit (baseline)', 'Model UNICODE 1_9', 'Model unfrozen 1_9' and 'Model 16_bit 1_9'.

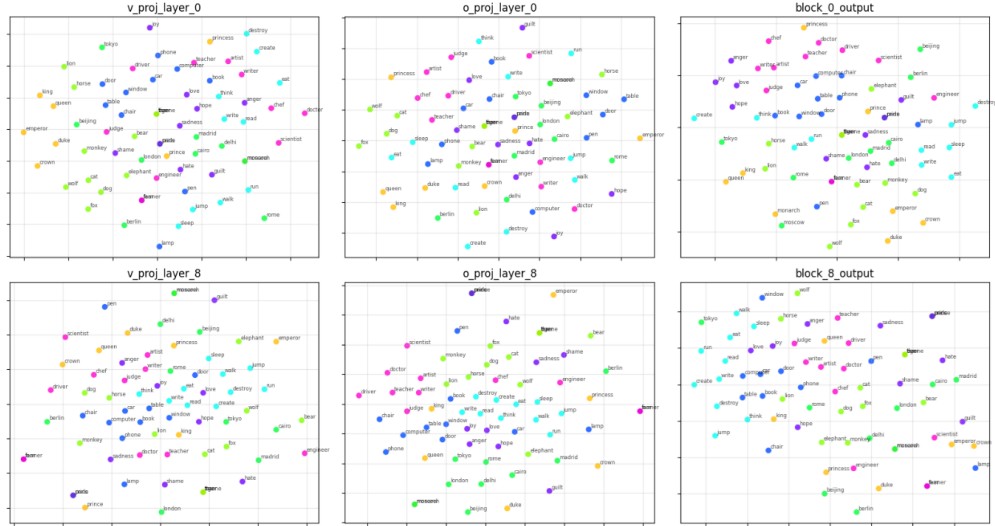

Figure 9: t-SNE visualization for the first and the last layer attention projections (v_proj and o_proj), and transformer block outputs for 'Model unfrozen (baseline)' with trainable embedding layer. In this picture semantic groups (e.g., numbers, professions, animals) tend to make clusters.

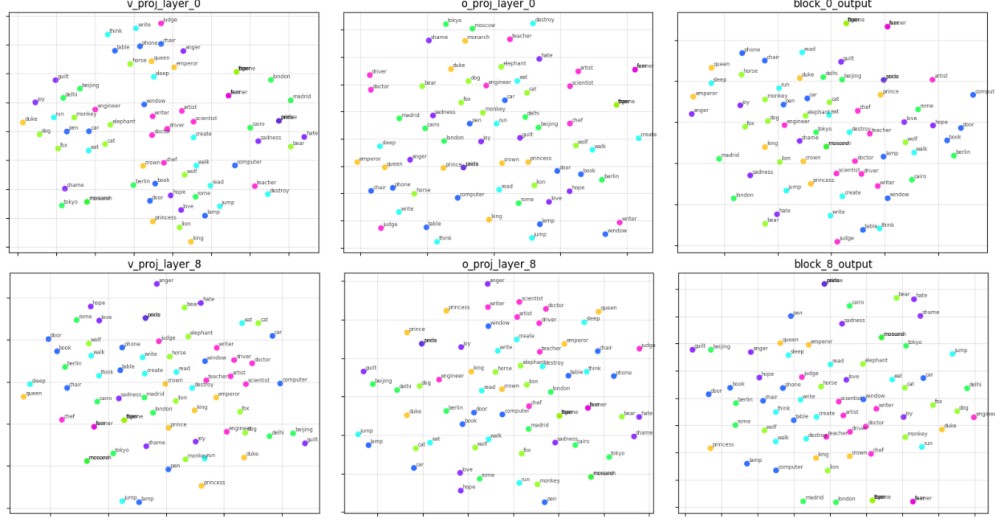

Figure 10: t-SNE visualization for the first and the last layer attention projections (v_proj and o_proj), and transformer block outputs for 'Model UNICODE 1_9' with frozen embedding layer. In this picture semantic groups (e.g., numbers, professions, animals) tend to make clusters **after** frozen embedding layer demonstrating emergence of semantic structure.

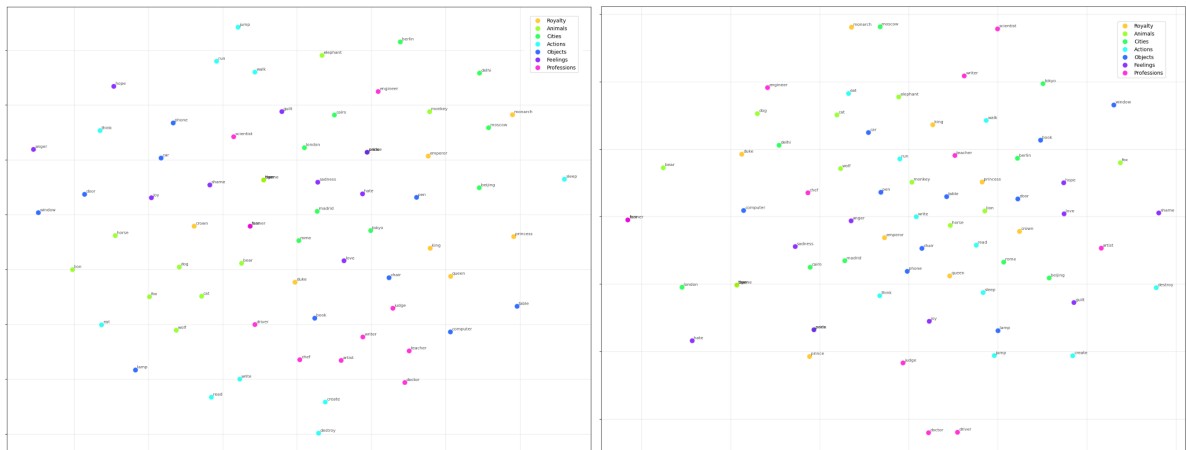

Figure 11: t-SNE visualization of frozen input token embeddings for 'Model unfrozen (baseline)' (clustering, left) and 'Model UNICODE 1_9' (no semantic clusters by design, right).

The ablation studies provide further crucial insights. The 'Model unfrozen 1_9', which is grown constructively but uses standard trainable embeddings, fails to match the performance of either the monolithic baseline or our proposed model on MMLU. This indicates that the constructive growth strategy alone is insufficient; the stability of the frozen substrate is a critical component for achieving superior reasoning. Remarkably, the 'Model 16_bit 1_9', despite using extremely impoverished 16-bit binary embeddings, still achieves competitive performance, underscoring the Transformer's powerful ability to induce semantic structure from even minimal, consistent input signals.

These performance differences are mirrored in the models' internal representations. Figure 11 confirms the initial state: the baseline's input embeddings are semantically clustered after training, while our model's are not semantically clustered by design. However, visualizations of the final layers (Figure )10) show that rich semantic clusters (e.g., numbers, professions) clearly emerge within the deeper layers of 'Model UNICODE 1_9', rivaling the structure seen in the baseline (Figure 9). This provides direct visual evidence that semantics are being constructed compositionally within the network, rather than being learned at the input layer.

In summary, this controlled study demonstrates that constructive growth on a frozen substrate is not only a viable training strategy but can lead to superior reasoning performance compared to traditional monolithic training, by allowing the model to more effectively specialize its deeper layers for semantic composition.

# 6 DISCUSSION

Our findings present a compelling case for a constructive approach to building LLMs, moving away from the monolithic paradigm.

## 6.1 The Frozen Embedding as a Universal Docking Port

The key enabler for our methods is the shared, frozen representational substrate. It acts as a universal standard, an "API" or "docking port" for neural components. When all models speak the same fundamental input/output language of visual forms, their higher-level learned knowledge (the transformations within their layers) becomes interoperable.

## 6.2 From Monolithic Forging to Constructive Growth

Standard LLM training is like trying to forge a complex machine from a single, molten block of metal—immensely difficult and inflexible. Our progressive growth method is more akin to building a

skyscraper floor by floor, or a living organism growing cell by cell. Each new layer builds upon a stable, functional foundation. This incremental process is more computationally tractable, more interpretable, and inherently more adaptable. The emergence of complex abilities like SQuAD performance only at significant depth is analogous to the development of higher-order cognitive functions in a growing brain, which require a sufficient hierarchy of neural processing.

### 6.3 Implications for a Modular AI Ecosystem

This paradigm has significant implications for the future of AI development:

- Resource Efficiency and Specialization: Organizations could train smaller, expert models on proprietary or specialized data.

- Continual Learning: New knowledge and skills can be added by training and adding new expert modules or by extending the core model with new layers, drastically reducing the risk of catastrophic forgetting.

- **Synergy with Model Quantization:** Our approach, particularly with the minimalist 16-dimensional binary embeddings, introduces a "quantization-native" input layer. Standard models learn continuous, high-precision embeddings that must be aggressively quantized post-training. In contrast, our method is built upon an inherently discrete foundation: the embedding vectors themselves are derived from binary, monochrome glyph images. This creates a natural synergy with quantization techniques, as the model is trained from the outset to compose meaning from low-precision, discrete inputs, rather than learning continuous representations that must be aggressively quantized post-hoc. This invites future research into whether a discrete representational substrate can facilitate more efficient and performant quantization of the entire model.

- Democratization: This approach lowers the barrier to entry. Instead of requiring the resources to train a 100 B+ parameter model from scratch, researchers could contribute to a larger ecosystem by developing and sharing smaller, compatible modules.

## 7 CONCLUSION

Building on the foundation of frozen visual embeddings, we have demonstrated novel and efficient paradigm for progressive layer-wise growth. This work reframes the challenge of scaling AI from a monolithic endeavor to a constructive and modular one. By establishing a fixed, universal representational substrate, we unlock a new design space for creating powerful, flexible, and efficient AI systems. This "constructive learning" paradigm offers a promising path toward a more sustainable and collaborative future for artificial intelligence.

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

# 8 APPENDIX A: Training loss and perplexity data

Table 1: Training loss and perplexity data

| Step | Model unfrozen (base line) Loss | Model unfrozen (base line) PPL | Model frozen UNI CODE (base line) Loss | Model frozen UNI CODE (base line) PPL | Model frozen 16_bit (base line) Loss | Model frozen 16_bit (base line) PPL | Model unfrozen 1_9 Loss | Model unfrozen 1_9 PPL | Model UNI CODE 1_9 Loss | Model UNI CODE 1_9 PPL | Model 16_bit 1_9 Loss | Model 16_bit 1_9 PPL |
|---|---|---|---|---|---|---|---|---|---|---|---|---|
| 0 | 11.34 | 84005 | 11.12 | 67363 | 11.40 | 89155 | 10.88 | 53254 | 10.67 | 43094 | 11.32 | 82104 |
| 050000 | 2.90 | 18.27 | 3.35 | 28.62 | 3.11 | 22.50 | 2.21 | 9.11 | 2.36 | 10.63 | 2.37 | 10.73 |
| 100000 | 2.29 | 9.82 | 2.47 | 11.85 | 2.38 | 10.85 | 1.72 | 5.61 | 1.76 | 5.79 | 1.84 | 6.29 |
| 150000 | 2.00 | 7.38 | 2.14 | 8.48 | 2.15 | 8.60 | 1.63 | 5.08 | 1.65 | 5.22 | 1.69 | 5.43 |
| 200000 | 1.86 | 6.45 | 2.00 | 7.35 | 2.06 | 7.85 | 1.59 | 4.93 | 1.61 | 5.03 | 1.66 | 5.24 |
| 250000 | 1.77 | 5.90 | 1.85 | 6.37 | 1.96 | 7.10 | 1.56 | 4.78 | 1.60 | 4.94 | 1.60 | 4.95 |
| 300000 | 1.71 | 5.52 | 1.75 | 5.76 | 1.84 | 6.28 | 1.53 | 4.62 | 1.55 | 4.73 | 1.57 | 4.83 |
| 350000 | 1.66 | 5.24 | 1.70 | 5.45 | 1.74 | 5.71 | 1.50 | 4.48 | 1.53 | 4.63 | 1.54 | 4.67 |
| 400000 | 1.60 | 4.98 | 1.63 | 5.11 | 1.66 | 5.26 | 1.47 | 4.37 | 1.51 | 4.51 | 1.52 | 4.58 |
| 450000 | 1.54 | 4.69 | 1.56 | 4.77 | 1.58 | 4.86 | 1.45 | 4.27 | 1.49 | 4.41 | 1.49 | 4.44 |
| 500000 | 1.50 | 4.48 | 1.50 | 4.49 | 1.53 | 4.63 | 1.44 | 4.22 | 1.47 | 4.33 | 1.46 | 4.32 |
| 550000 | 1.46 | 4.29 | 1.47 | 4.34 | 1.49 | 4.42 | 1.42 | 4.12 | 1.44 | 4.22 | 1.44 | 4.22 |
| 600000 | 1.40 | 4.06 | 1.40 | 4.04 | 1.43 | 4.17 | 1.42 | 4.12 | 1.44 | 4.21 | 1.43 | 4.19 |
| 650000 | 1.38 | 3.96 | 1.39 | 4.02 | 1.39 | 4.03 | 2.25 | 9.45 | 2.02 | 7.53 | 2.02 | 7.54 |
| 700000 | 1.35 | 3.87 | 1.37 | 3.92 | 1.37 | 3.93 | 1.41 | 4.09 | 1.43 | 4.18 | 1.42 | 4.14 |
| 750000 | 1.33 | 3.78 | 1.34 | 3.80 | 1.35 | 3.85 | 1.41 | 4.09 | 1.43 | 4.17 | 1.42 | 4.15 |
| 800000 | 1.31 | 3.71 | 1.32 | 3.75 | 1.35 | 3.86 | 1.40 | 4.06 | 1.42 | 4.14 | 1.42 | 4.16 |
| 850000 | 1.33 | 3.77 | 1.31 | 3.71 | 1.31 | 3.72 | 1.40 | 4.04 | 1.42 | 4.12 | 1.43 | 4.16 |

# 9 APPENDIX B: Parameter Details for Large-Scale Model Growth

Table 2 provides a detailed breakdown of the parameter counts for the large-scale model growth experiment. The base architecture is $d_{\text{model}} = 4096$, $n_{\text{head}} = 32$, and a vocabulary size of 131,072.

A key aspect of our methodology is maintaining a constant budget of trainable parameters at each stage, constrained to approximately 740M to fit within the memory of a single H100 accelerator. This budget is allocated in two strategic ways:

- **Layer Growth:** In the primary growth stages, the budget is dedicated entirely to training the parameters of a newly added top layer, while all preceding layers remain frozen.

- **Holistic LoRA Tuning:** At later stages (e.g., after 3 and 5 layers), the same budget is reallocated to train low-rank adapters applied to *all* layers simultaneously. The LoRA rank and target modules were chosen to ensure the total number of trainable adapter parameters also fit within the $\approx$740M

budget. This allows for a global, computationally efficient fine-tuning of the entire model stack to better integrate the new layers without increasing the instantaneous memory footprint.

Table 2: Layer-wise growth parameters for the large-scale (ABS) model. Each training stage operates under a fixed budget of ≈740M trainable parameters. This budget is allocated either to a new top layer (Method: Growth) or to LoRA adapters across the entire stack (Method: Fine-tuning).

| Model Stage | Layers | Method | Total Params (M) | Trainable (M) | Frozen (M) |
|---|---|---|---|---|---|
| ABS-1 | 1 | Growth | 1275.2 | 738.4 | 536.9 |
| ABS-2 | 2 | Growth | 1476.6 | 738.4 | 738.2 |
| ABS-3 | 3 | Growth | 1677.9 | 738.4 | 939.5 |
| ABS-3 (LoRA) | 3 | Fine-tuning | 1728.3 | 503.3 | 1225.0 |
| ABS-4 | 4 | Growth | 1879.3 | 738.4 | 1140.9 |
| ABS-5 | 5 | Growth | 2080.7 | 738.4 | 1342.3 |
| ABS-5 (LoRA) | 5 | Fine-tuning | 2080.7 | 731.3 | 1349.4 |
| ABS-6 | 6 | Growth | 2282.0 | 738.4 | 1543.6 |
| ABS-6 (LoRA) | 6 | Fine-tuning | 3008.7 | 726.6 | 2282.1 |

## 10 APPENDIX C: Parameter Details for Controlled Study Models

Table 3 provides the corresponding parameter breakdown for the smaller-scale models used in our controlled comparative study. The base architecture for these models is $d_{model} = 1024$, $n_{head} = 32$, and a vocabulary size of 65,536. The models were grown in stages of three layers each, with previously trained layers being frozen at subsequent stages.

Table 3: Parameter breakdown for the controlled study models (final size ≈248M).

| Model Type | Stage (Layers) | Total (M) | Trainable (M) | Frozen (M) |
|---|---|---|---|---|
| Constructive (UNICODE) *(Our Method)* | 1-3 | 172.1 | 105.0 | 67.1 |
| | 1-6 | 209.8 | 37.7 | 172.1 |
| | **1-9 (Final)** | **247.6** | **37.7** | **209.9** |
| Constructive (Trainable Emb.) *(Constructive trainable)* | 1-3 | 172.1 | 172.1 | 0.0 |
| | 1-6 | 209.8 | 37.7 | 172.1 |
| | **1-9 (Final)** | **247.6** | **37.7** | **209.9** |
| Constructive (16-bit Emb.) *(Ablation)* | 1-3 | 106.0 | 105.0 | 1.0 |
| | 1-6 | 143.8 | 37.7 | 106.1 |
| | **1-9 (Final)** | **181.6** | **37.7** | **143.9** |
| Monolithic (Trainable Emb.) *(Baseline trainable classic)* | **1-9 (Final)** | **247.6** | **247.6** | **0.0** |
| Monolithic (Frozen UNICODE Emb.) *(Baseline)* | **1-9 (Final)** | **247.6** | **180.5** | **67.1** |
| Monolithic (Frozen 16_bit Emb.) *(Baseline)* | **1-9 (Final)** | **181.6** | **180.6** | **1.0** |

## 11 APPENDIX D: Training Hyperparameters

All models in the controlled comparative study were trained with the same set of hyperparameters to ensure a fair comparison. The key parameters are listed in Table 4. We did not perform extensive hyperparameter tuning; these values were chosen based on standard practices for models of this scale.

Table 4: Key training hyperparameters for the controlled study.

| Hyperparameter | Value |
| --- | --- |
| Optimizer | AdamW |
| Learning Rate | 1.8 * 1e-3 |
| Batch Size | 36 |
| Block Size | 1024 |
| Grad Accum | 125 |
| Weight Decay | 0.1 |
| Adam $\beta_1$ | 0.9 |
| Adam $\beta_2$ | 0.95 |
| Gradient Clipping | 0.1 |
| Warmup Steps | 2000 |

