# OpenReview forum: "Growing Transformers: Modular Composition and Layer-wise Expansion on a Frozen Substrate"
_TMLR — Rejected by TMLR_

### Review · Reviewer_jc4Q · 2025-08-20

**Summary Of Contributions:**

In this work, the authors study two problems: merging models and growing models one component at a time (in depth). They use models with fixed embeddings in this work, motivated by the idea that shared initial representations make models more compatible.

In the first set of experiments, the authors combined an english+russian model with an english+chinese model by simply averaging their logits together. They claimed that this improved performance on a variety of tasks.

Next, they performed an experiment where they trained a transformer on MMLU with the following procedure: first they trained a one layer transformer. Next, they added a new layer, randomly initialized, and trained only this layer. They repeated this procedure until they ended up with a 6 layer deep transformer model. They showed that the loss is generally decreasing with the number of layers, with loss spikes when the new model is first initialized.

Overall this work suffers from the fact that there are not good baseline comparisons; it also to my knowledge does not have any unique contributions as there have been more detailed studies of both MOE strategies as well as model growth techniques.

**Audience:**

No

**Audience Explanation:**

MOE models are well-studied, and indeed are now industry standard [1, 2]. Indeed, some of the earliest works exploiting MOEs at scale were in the multilingual setting [3]. Readers will not be surprised to learn that MOEs can improve performance; the simple "average MOE" is less expressive than MOE models studied more generally, and indeed is a special case of some of those models. In usual applications, training the routing function is not the limiting factor in MOE deployment. Therefore the MOE part of this paper is of little interest to TMLR readership.

With regards to the model growing: other works have asked this question in more detail and with more rigor, see for example [4]. It is well-established that increasing model size, even in this iterative way leads to better performance than smaller models; the open questions center around whether or not this can be done in a way that is more computationally efficient than the current paradigm. This paper makes no progress towards that latter question

[1] https://arxiv.org/abs/2412.19437

[2] https://arxiv.org/abs/2401.04088

[3] https://arxiv.org/abs/2006.16668

[4] https://proceedings.mlr.press/v162/shen22f.html

**Claims And Evidence:**

No

**Claims Explanation:**

The writeup of the experiments is very unclear. In section 3.1 it is implied that the MOE model approach requires no further training. However section 4.1 shows training curves. In light of this it is not clear to me what experiments were actually done. This also then brings into question the claim that the MOE is the best model, since there appears to be additional training that may or may not have been done for the other models. Indeed the MOE model is not the best on any single task except final_arc_e, which makes claims of its superiority questionable as well.

With regards to the depth growing experiments, there is a key baseline missing: comparison against a model where multiple layers were trained at once. It is not surprising at all that larger models can lead to improved results; the question is whether or not a particular growth or scaling strategy outperforms another.

**Requested Changes:**

Crucial issues to address:

What is the actual training procedure in the experiments? Particularly those in 4.1?

Is there some quantitative metric that shows the MOE model in section 4.1 is actually better than the others, when compared fairly in terms of training and inference budget?

How does the performance of the grown model compare to a non-grown, deeper model with similar training budget?

Even with these issues resolved I may not recommend acceptance, see answer to "Would at least some individuals in TMLR's audience be interested in knowing the findings of this paper?" above.

---

> ### Author Response · Authors · 2025-10-06
>
> We thank Reviewer for sharp and highly accurate feedback.
>
> The reviewer's central criticism—the absence of a baseline comparing against a monolithically trained model—was spot on and has become the cornerstone of our revision.
>
>
> Regarding the Missing Baseline for Growth Experiments: We agree that this was a critical flaw. We have designed and executed a new controlled comparative study to explicitly address this. We now directly compare the performance of our 9-layer constructively grown model against a 9-layer model of the identical architecture trained traditionally (end-to-end). This new experiment directly answers the question of how our growth strategy compares to the standard paradigm.
>
>
> Regarding Unclear Training Procedures and MoE: We acknowledge the confusion in the original manuscript. To improve clarity, we have removed the MoE section entirely. The paper now focuses solely on progressive growth. We have added a new, detailed "Experimental Setup" section that precisely describes the architectures, baselines, and training protocols for all experiments, leaving no ambiguity.
>
>
> Regarding Novelty and Computational Efficiency: The reviewer rightly points out that the key question is efficiency. Our revision reframes the contribution in this light. The large-scale experiment is presented as a method for training models under memory constraints (i.e., when optimizer states for the full model do not fit on GPUs).
>
> The controlled study then shows that this resource-efficient method does not come at a performance cost for reasoning tasks, and can even offer benefits (as seen on MMLU).
>
> While we do not claim to have fully solved the computational efficiency question, we believe our work, especially with the new baseline comparison, makes meaningful progress by demonstrating a viable and effective alternative to monolithic training.

---

> > ### Comment · Reviewer_jc4Q · 2025-10-10
> > **Response to revision**
> >
> > I thank the authors for providing a revised version of the text; I do believe that the revision is an improvement and the paper is generally more readable.
> >
> > While I do think there are some interesting results in the comparison between the different model growth strategies, the overall message is still a bit unclear to me. The relationships to other works in the area are still not explored properly. In addition the training details still do not appear in the paper (e.g. hyperparameter tuning setup).
> >
> > The main effect being probed by the studies is the usefulness of the fixed embedding, and not anything in particular about the growth strategy. The key experiment that is missing is training a model with the fixed embedding, with all other training done normally.
> >
> > Given that this is such a major revision, I think the presentation needs work; in particular I'm not sure what I'm supposed to gain from all the t-SNE visualizations. Also it seems that Arc-c is listed twice, once in Figure 7 and once in Figure 8.
> >
> > Overall while the authors are moving in the right direction, I think this paper needs some more time to develop before I can recommend acceptance.

---

> > > ### Author Response · Authors · 2025-10-13
> > >
> > > We sincerely thank the reviewer for the clear and actionable feedback on our revision. We are very grateful for this guidance.
> > >
> > > The reviewer's central point is exceptionally well-taken: the key missing experiment is a monolithically trained model with fixed embeddings. We agree completely that this is necessary to disentangle the core effects.
> > >
> > > We have already begun running this experiment and are also simplifying the t-SNE visualizations and adding a hyperparameter table as requested.
> > >
> > > We anticipate submitting a revised manuscript that incorporates these results within 3-5 days.

---

> > > ### Author Response · Authors · 2025-10-16
> > >
> > > We thank you for your review of our revised manuscript.
> > >
> > >
> > > Your central point about the missing baseline was exceptionally well-taken. We agree completely that this was necessary to disentangle the effects of our methodology.
> > >
> > >
> > > We have now conducted these crucial experiments, and we believe the new results provide the clarity you requested.
> > > Here is how we addressed your final concerns point-by-point:
> > >
> > >
> > > Regarding the key missing experiment: We have now trained two such monolithic baselines: one with our standard frozen Unicode embeddings (Model frozen UNICODE (baseline)) and another with the minimalist n_embed=16 binary frozen embeddings (Model frozen 16-bit (baseline)).
> > >
> > >
> > > These new baselines are now central to our Controlled Comparative Study (Section 4.2.2). The results, presented allow us to directly compare constructive growth vs. monolithic training on an identical frozen substrate. This helps to isolate the specific contribution of the growth strategy itself, separate from the effect of the frozen embeddings.
> > >
> > >
> > > Regarding the overall message (usefulness of fixed embedding vs. growth strategy):
> > >
> > > Our revised narrative and new experiments now clarify that our core hypothesis involves the synergy between these two components. Our results indicate that while the frozen substrate is crucial (as Constructive Trainable underperforms), the growth strategy on this substrate provides a distinct advantage on reasoning tasks like MMLU compared to monolithic training on the same frozen substrate.
> > >
> > >
> > > Regarding missing training details:
> > >
> > > We have now added Appendix D: Training Hyperparameters, which contains a comprehensive table listing all key hyperparameters used in the controlled study to ensure a fair comparison and improve reproducibility.
> > >
> > >
> > > Regarding the t-SNE visualizations:
> > >
> > > We have revised them to tell a simpler, more direct story.

---

### Review · Reviewer_EgPe · 2025-08-26

**Summary Of Contributions:**

## Strength
1. Concrete words.
2. Beautiful graph.

## Weakness
1. Poor Writing:
* In 3.1, Logit averaging and adapter-based fusion are mentioned. However, in 4.1, I don't which part does the experiment corresponds to.
* How do you define loss in Sec 4.1? The elements of Fig 1, 2, 3 aren't clearly written, such as, what is best_bvv_unfrozen_zh?
2. Unclear Motivation:
* Why should we care about whether the input representation is fixed? I don't understand why we need to fix the representation when growing the network horizotally?
3. Lack of related works: I believe that works of neural architecture search are strongly related to the paper, but they aren't discussed. Also, works of model merging aren't fully discussed (such as lacking https://arxiv.org/abs/2408.07666, https://arxiv.org/abs/2505.12082 and so on)
4. Lack of insights: The experiments are extremely rough and the analysis is unclear and even wrong. For example, the author claims that "For instance, 'best_bvv_moe' ... close to the final loss of expert", "This confirms that the composition immediately ... without catastrophic interference". The logic is incorrect. Why the loss can reflect this? You calculate the L2 distance between weights.
5. Lack of experiments: Experiments are only done in limited domains, restricting the generalizability of their claims.

**Audience:**

No

**Audience Explanation:**

The claims aren't well justified.

**Broader Impact Concerns:**

No.

**Claims And Evidence:**

No

**Claims Explanation:**

For Seamless Model Composition, only 1 training dataset is used, and the components of the training dataset aren't clearly written.
For layer growth, they only test their claim to a layer number of 3. For layer numbers greater than 3, they need lora to tune preceding layers. Furthermore, the largest layer number is 6. This limitations of the experiments can support their claims

**Requested Changes:**

Please see weakness part.

---

> ### Author Response · Authors · 2025-10-06
>
> We thank Reviewer for the detailed and concrete feedback.
>
> The comments highlighted several areas where our writing and experimental design were unclear. We have thoroughly revised the manuscript to address these weaknesses.
>
>
> Regarding Poor Writing and Unclear Experiments: We agree that the initial submission was confusing, particularly the section on MoE. To resolve this, we have completely removed the MoE experiments to focus the paper entirely on the clearer, more novel story of progressive layer-wise growth. We have rewritten the experimental sections to be far more explicit, with a clear description of the two new experimental setups (feasibility and controlled comparison) and detailed tables in the Appendix.
>
>
> Regarding Unclear Motivation: We have revised the Introduction to clarify the motivation. We posit that because semantics are emergent, lower layers can be frozen, which makes incremental, resource-efficient growth a viable scaling strategy. This is now the central thesis of the paper.
>
>
> Regarding Lack of Related Works: We have removed the MoE section, making the model merging references less relevant. However, we have retained and contextualized our work with respect to greedy layer-wise training and progressive architectures.
>
>
> Regarding Lack of Insights and Experiments: This was a critical point. To provide deeper insights, we have added t-SNE visualizations that directly show the emergence of semantic clusters in deep layers.
>
> To address the lack of experiments, we have introduced a new controlled comparative study with a crucial monolithic baseline and two other ablations, providing a much more rigorous validation of our claims. We now test our claim not just up to 6 layers, but on a complete 9-layer model compared directly with a traditionally trained counterpart.

---

> > ### Comment · Reviewer_EgPe · 2025-10-15
> > **Response the authors**
> >
> > Thanks to the authors for the response and their efforts in making the paper better! However, I hope the authors can make a direct response to my proposed questions, such as my claim of unsolid statements, definition of the loss function. Thanks so much!

---

> > > ### Author Response · Authors · 2025-10-16
> > >
> > > We sincerely thank the reviewer for this follow-up and for holding us to a high standard.
> > >
> > >
> > > We apologize for our oversight in not addressing your original points directly in the first round of revisions.
> > > Our reasoning was that by removing the MoE section, we were implicitly acting on your feedback,
> > > but we see now that this was insufficient and left your questions unanswered.
> > >
> > >
> > > To correct this, here are the direct answers to your original points regarding the now-removed MoE section:
> > >
> > >
> > > Regarding "unsolid statements" and "Why the loss can reflect this":
> > >
> > > Our original statement about the low starting loss was based on an informal observation (merging models with trainable embeddings led to a significantly higher initial loss in our tests), but we agree this observation is not rigorous proof of "no catastrophic interference." Your critique on this point was a key reason we felt the entire MoE topic required a separate, more focused study, leading us to remove it to concentrate on the more rigorously tested layer-growth claims.
> > >
> > >
> > > Regarding the definition of the loss function:
> > >
> > > In the original MoE experiment, the loss was the standard cross-entropy, calculated on the element-wise averaged logits from the two experts:
> > > loss = F.cross_entropy(averaged_logits, targets).
> > >
> > >
> > > Regarding "Unclear Motivation" for fixed embeddings in MoE:
> > >
> > > The motivation was precisely that a shared, frozen embedding layer could act as a stable "docking port," allowing independently trained experts to be merged more effectively than models with disparate, trainable embeddings.
> > > Our preliminary (but not rigorous) finding of a lower starting loss supported this.
> > >
> > >
> > > However, we agree with you that proving this hypothesis requires a much more extensive set of experiments and baselines.
> > >
> > >
> > > Regarding "Lack of experiments, insights, and related works" for MoE: Your points on these weaknesses were instrumental in our decision to revise the paper.
> > >
> > >
> > > Thank you again for your guidance.

---

### Review · Reviewer_SmxR · 2025-09-23

**Summary Of Contributions:**

The paper investigates two techniques, based on an earlier observation that Transformers model can achieve strong performance even on a frozen embedding layers based on unicode characters. This suggests that models can be trained over previously trained frozen components, which lead to the two experiments in this paper:
- post-hoc merging of models (MoE)
- progressive layer-wise growth.

The paper provides perplexity scores and MMLU, SQuaD benchmarks to support its claims.

**Audience:**

No

**Audience Explanation:**

I am on the fence on this. If there were a "Maybe" button I might have clicked that. While the paper does have an interesting thesis (the unicode embeddings), because the results are so far from SOTA, I cant say for sure how convincing it will be to today's readers.

**Claims And Evidence:**

No

**Claims Explanation:**

In my honest opinion the results of the paper were not very convincing to me for the following reasons:
1. the paper focuses on benchmarks and architectures that are, as the paper admits, not SOTA, and uses only a modest number of 9B tokens for training. Hence, it is hard for me to envision how the results can extend to more sophisticated benchmarks.
2. the unicode embedding layer is very interesting, and if it has applications to models today that is great. But again from the results it is hard for me to envision this.

Granted there are some novel differences, such as Section 4.1 differs from model merging by performing logit averaging or fusion, and Section 4.2 differs in the sense that the model is grown from scratch, it does not seem very convincing in that it brings the following questinos.
- Figure 1 shows low loss and PPL for the  best_bvv_moe, however this is a limited example . There is a similar trend shown in Figure 2 for the max_bvv_moe. Unfortunately loss and PPL comparisons for me are not very convincing. In Figure 3 the numbers look on the low side.
- Figure 4 again shows loss and PPL numbers with the ARC benchmark (again numbers look on the low side).  Figure 5 looked interesting, but the "jump" is again not convincing to me because the SqUAD benchmark is again on the low side.


And finally, do these observations carry on to

**Requested Changes:**

While I understand that the aim of the paper is not to pursue SOTA results, I feel that in this age it is inevitable to generate interest from the community.
- Need to have very strong proof why training over frozen layers can work, because this is an idea that has been played with by the community for awhile. If such an idea works because of the unicode embeddings, then some very strong proof needs to be shown. It is often thought in the community that freezing part of the model can be suboptimal (e.g., see https://arxiv.org/abs/2405.09673), so is there some very strong proof or intuition why this does not need to be the case (apart from just showing PPL results). Is there some probing that can be done to show clear emergence of reasoning patterns?
- Need a convincing application for MoE / progressive growth. While today's models are pushing the limits, where a 7B model is considered not too big, what are the applications for training a model layer by layer? Is this for training models on memory limited hardware, to save on optimizer parameters? If we slowly unfreeze parameters during training, does it affect the overall convergence speed (I suspect it might), so then what are the clear use cases for a method like this?
- Consider adding some generation benchmarks with skills like math, code, and show that your method is applicable in these domains.

---

> ### Author Response · Authors · 2025-10-06
>
> We thank Reviewer for valuable feedback.
>
> The comments about the lack of convincing proof and the need to compare against more standard training paradigms were particularly insightful. We have undertaken a major revision to address these points.
>
>
> Regarding the need for "very strong proof" and comparison to suboptimal freezing: We completely agree that perplexity results alone are not sufficient. The core of our revised manuscript is a new controlled comparative study where we benchmark our constructive growth method directly against a monolithically trained baseline of the same size.
>
> This directly addresses the question of whether our method is suboptimal. The results show that our method outperforms the monolithic baseline on the MMLU reasoning benchmark, providing the strong, non-PPL evidence requested. Furthermore, we've added t-SNE visualizations to offer a probing analysis, showing the clear emergence of reasoning patterns in the deep layers.
>
>
> Regarding the motivation and application for progressive growth: We have revised the Introduction and added section to explicitly state the motivation. We frame it in two ways: (a) a practical method for training models that are too large to fit into a single accelerator's memory during monolithic training (our 2.3B parameter experiment demonstrates this), and (b) a scientific tool to study the emergence of capabilities with depth. We have also added detailed tables in the Appendix (e.g., Appendix B) showing how the training budget is managed, addressing the question of memory savings.
>
>
> Regarding SOTA results and benchmarks: We acknowledge that our models are not SOTA. We have clarified in the revised paper that our primary goal is a controlled scientific study to isolate the effects of a training methodology, for which smaller-scale, reproducible experiments are more suitable. While we agree that math/code benchmarks would be valuable, we believe the current, more focused study on reasoning (MMLU, SQuAD) provides a strong and clear initial validation of the core idea.

---

> > ### Comment · Reviewer_SmxR · 2025-10-10
> > **Response to Author**
> >
> > Hi thanks for the improved manuscript. Definitely removing the discussion on MoE, and focusing on the layer growth-strategy, has greatly improved the presentation. Here are my thoughts after reading through the revision:
> >
> > Improvements
> > 1. The study in 4.4.2 is good; the choice of ablations is good. In particular the training dynamics in Figs 6-8 is quite illuminating of the growth approach; in Fig 7 the improvement of the proposed strategy for MMLU is nice. It was also nice to see that once a new layer is added, the performance recovers to a higher level, suggesting that the freezing does not compromise the generalization.
> >
> > Questions that still remain in my head:
> > 1. Even though in 4.2 the authors mentioned that the architecture is decoder, due to the size of the model, the reported benchmark performances in Fig 7 and Fig 8 are still rather low. This may dilute the contributions and challenge the proposed method if it could scale for larger models.
> > 2. In the Arc-c and C-sense training dynamics, it is very hard to see what is the benefit of the proposed method, since the performances are essentially on top of each other (or in fact it got worse in Fig 8 - top). This may dilute the contributions and challenge the success of the proposed method that it only works for benches similar to MMLU.
> > 3. While I thank the authors for the effort to perform the probing, the FIgures 4-5 are very hard to read and did not convey the main point clearly to me. When a reader looks at a plot, it is hard to focus on so many things. It is best to have a plot that convey simple strong points in a clear manner.

---

> > > ### Author Response · Authors · 2025-10-13
> > >
> > > We sincerely thank the reviewer for the very helpful feedback.
> > >
> > > Regarding the task-specific benefit (MMLU vs. ARC/C-SENSE):  We agree that the benefit is currently most pronounced on MMLU. In our final analysis, we will explicitly acknowledge this task-dependent nature. We believe the addition of one final baseline experiment (a monolithically trained model with frozen embeddings, as suggested by another reviewer) will help us further dissect this effect and discuss why certain reasoning tasks may benefit more from our approach.
> > >
> > > Regarding the low absolute scores and scalability: We acknowledge this limitation. Our goal with this study is a rigorous, controlled comparison of training methodologies, for which smaller-scale models are more feasible. In the final version, we will further clarify that our contribution lies in the relative performance gains and the novel training strategy, which we believe provides a foundation for future scaling experiments.
> > >
> > > We have already begun this work and expect to upload a final, improved manuscript within 3-5 days.

---

> > > ### Author Response · Authors · 2025-10-16
> > >
> > > Thank you for your positive and encouraging feedback on our revised manuscript.
> > >
> > >
> > > Regarding low absolute scores and scalability:
> > >
> > > We agree that the absolute scores are not state-of-the-art and acknowledge this limitation directly in the revised manuscript (Section 1). Our primary goal with this work was not to compete for SOTA, but to conduct a rigorous, controlled comparison of training methodologies. For such a scientific study, smaller-scale models allow for the necessary ablations and baselines to be run under controlled conditions. We believe the significant relative performance gain of our method on MMLU provides a strong proof-of-concept for the paradigm, which we hope will inspire future large-scale experiments by the community.
> > >
> > >
> > > Regarding the task-specific benefit (MMLU vs. ARC/C-SENSE):
> > >
> > >
> > > We hypothesize that MMLU, which often requires more abstract, multi-step reasoning, benefits disproportionately from our method's decoupling of low-level structural representation (in the frozen layers) from high-level semantic composition (in the trained layers). In contrast, tasks that may rely more on retrieving specific factual knowledge (like ARC) show less of a gap. We believe this is an interesting finding in itself and now frame it as such.
> > >
> > >
> > > Regarding the clarity of the t-SNE visualizations:
> > >
> > >
> > > We have simplified the visualizations and improved their captions:
> > >
> > > (1) The input frozen embeddings are intentionally non-semantic.
> > >
> > > (2) Rich semantic clusters emerge and sharpen as we go deeper into the model.
> > >
> > > (3) This process of emergence is visually confirmed to happen in our constructively grown model, proving that meaning is being built by the network's layers.

---

### Author Response · Authors · 2025-10-06

Dear Action Editor and Reviewers,

We sincerely thank the Action Editor and all three reviewers for their time and insightful, constructive feedback. The reviews made it clear that our initial submission suffered from a lack of focus and, most critically, the absence of a proper baseline to evaluate the core claim of constructive growth.

Taking this crucial feedback to heart, we have performed a major revision of the manuscript. The key changes are:

Refocused Narrative: We have completely removed the section on Mixture-of-Experts (MoE) to concentrate entirely on the more novel and central claim of the paper: the viability and effectiveness of progressive layer-wise growth. This addresses the reviewers' concerns about lack of novelty in the MoE section and clarifies the paper's contribution.

Introduction of a Critical Baseline Study: The core of our revision is a new, rigorous controlled comparative study (Section 5.3). To directly address the main criticism from all reviewers, we now compare our constructive growth method against a monolithically trained Transformer of the exact same final architecture and size. This provides the missing baseline.

Strengthened Motivation and Ablation Studies: We have clarified the motivation for this work (Section 1), framing it as a resource-efficient scaling strategy and a method for scientific inquiry. The new controlled study includes two further ablations: a constructive model with trainable embeddings, and one with minimalist 16-dimensional binary vector frozen embeddings, to isolate the effects of the growth strategy and the frozen substrate.


Deeper Analysis with Visualizations: To provide the "stronger proof" and "deeper insights" requested, we have added t-SNE visualizations of the models' internal representations (Figures 4,5,9,10,11). These plots provide direct visual evidence of how semantic structures emerge in the deeper layers of our constructively grown model.


Comprehensive Appendices: We have added detailed appendices with tables breaking down the parameter counts for all experiments (Appendix B, C) and raw training data (Appendix A) to improve clarity and reproducibility.


We believe these extensive revisions directly address the reviewers' primary concerns. We are grateful for the opportunity to improve our work based on the reviewers' guidance.

---

### Author Response · Authors · 2025-10-16

Dear Action Editor and Reviewers,


We are deeply grateful for your detailed and insightful feedback on our revision.


In this revision, we have incorporated the following key changes:


Addition of Two Crucial Monolithic Baselines: We have run the key missing experiments. We now include two new baselines: a monolithically trained model with frozen Unicode embeddings and another with frozen '16-bit' n_embed=16 binary embeddings. These additions are central to the new manuscript (Section 4.2.2), as they allow us to cleanly disentangle the effects of the constructive growth strategy from the effects of both rich (Unicode) and minimal (16-bit) frozen embedding substrates.


Simplified and Clarified Visualizations: We have redesigned the t-SNE visualizations. They are now simpler, with clearer titles and captions, each designed to convey a single, strong point about the emergence of semantic structure.


Strengthened Future Impact with Quantization Synergy: To address the broader impact and motivation, we have added a new discussion point (Section 6.3) that explores the natural synergy between our discrete, low-precision embedding approach and model quantization. This highlights a promising new direction for future research and a practical application of our findings.


Full Transparency on Training Details: We have added a new Appendix D with a comprehensive table of all training hyperparameters used in the controlled study to ensure full reproducibility.

---

### Decision · Action_Editor_Xz7U · 2025-10-30

**Recommendation:** Reject

**Additional Comments:**

This paper proposes a layer-wise constructive methodology for training language models by combining early-stage layer freezing with progressive fine-tuning of the full model stack as complexity increases. To validate the approach, the authors trained several smaller-scale models and evaluated them on a few standard benchmarks.

All reviewers recommended rejection, raising the following main concerns:

1.	In the original submission, reviewers expressed concerns about the MoE section—specifically its unclear motivation and inconsistent statements. Instead of providing further justification or clarification, the authors removed the entire section and shifted the paper’s focus to a different storyline. This change raises doubts about the overall soundness and coherence of the proposed approach, at least in its current form.

2.	The results are inconsistent: some benchmarks (e.g., MMLU) show improvement, while others (e.g., ARC and C-SENSE) do not, with no clear explanation provided.

3.	Several result plots (Figures 4, 5, 9, and 10) are difficult to interpret, and the paper lacks clear analysis or discussion of the data.

4.	Overall, the benchmark performance is weak and unconvincing.

5.	The models trained are small in scale.

6.	The paper provides little insight, either in experimental design or in the interpretation of results.

During the rebuttal phase, in addition to changing the storyline, the authors offered some clarifications. However, many experimental details remain missing. Therefore, this paper is recommended for rejection.

**Audience:**

No

**Audience Explanation:**

Although the proposed idea may have some potential, but the whole paper is not convincing in the current form.

**Claims And Evidence:**

No

**Claims Explanation:**

The evaluation results are not convincing to justify the methodology.